# Traffic Simulation Analysis on Running Speed in a Connected Vehicles Environment

**DOI:** 10.3390/ijerph16224373

**Published:** 2019-11-08

**Authors:** Bin Yu, Miyi Wu, Shuyi Wang, Wen Zhou

**Affiliations:** School of Transportation, Southeast University, Nanjing 211189, China; seuwmy@seu.edu.cn (M.W.); sywangseuer@seu.edu.cn (S.W.); 220193139@seu.edu.cn (W.Z.)

**Keywords:** connected vehicles, optimization speed model, VISSIM, running speed

## Abstract

Connected vehicles (CVs) exchange a variety of information instantly with surrounding vehicles and traffic facilities, which could smooth traffic flow significantly. The objective of this paper is to analyze the effect of CVs on running speed. This study compared the delay time, travel time, and running speed in the normal and the connected states, respectively, through VISSIM (a traffic simulation software developed by PTV company in German). The optimization speed model was established to simulate the decision-makings of CVs in MATLAB, considering the parameters of vehicle distance, average speed, and acceleration, etc. After the simulation, the vehicle information including speed, travel time, and delay time under the normal and the connected states were compared and evaluated, and the influence of different CV rates on the results was analyzed. In a two-lane arterial road, running speed in the connected state increase by 4 km/h, and the total travel time and delay time decrease by 5.34% and 16.76%, respectively, compared to those in the normal state. The optimal CV market penetration rate related to running speed and delay time is 60%. This simulation-based study applies user-defined lane change and lateral behavior rules, and takes different CV rates into consideration, which is more reliable and practical to estimate the impact of CV on road traffic characteristics.

## 1. Introduction

Connected vehicles (CVs) can exchange a variety of information including speed, acceleration, direction, and location instantly with nearby CVs and connected infrastructures. With real-time information, the forthcoming driving behavior could be predicted. Based on that, computers or human drivers could make critical driving decisions to minimize collision damages or avoid potential collisions [1]. CVs, with dual roles of exchanging information and decision-making, could improve the road capacity by enhancing the running speed, and thus contribute to a safer, more efficient, and comfortable driving experience [2].

Currently, studies on CVs mainly focus on three key aspects, environment perception, behavior decision, and motion control [3]. 

Environmental perception technology mainly relies on sensors. As regard to the latest technologies, laser radar can receive 60% to 75% of the three-dimensional information of vehicles’ surrounding environment [4], followed by camera’s visual information, for example CCD (Charge Coupled Device) camera millimeter wave radar’s direction and distance information, GPS’s position information, ultrasonic, infrared, and other photoelectric sensors’ information. Behavior decision systems, through which road and vehicle information can be obtained, help autonomous vehicles (AVs) drive more safely and reasonably. Motion control system is also used to control a vehicle itself [5]. Kang and Wang et al. proposed the limitations of current simulation methods: There is no prediction of complex relations between vehicles, driving environments, and routes [6]. They used a simulation model to build a series of methods on coordination, positioning, route planning, and evaluation for autonomous vehicles (AVs). Chen and Gong et al. pointed out that, in the complex and changeable dynamic environment, the acquisition of driving rules and decisions is still the key point restricting the development of AVs [7]. “Rough set theory” is an important method to resolve the issue. The motion control of AVs, including steering control, vertical and horizontal control, path tracking control, etc., is often inseparable from environmental perception [8]. Sense of environment allows vehicles to be controlled with fewer errors. Scholars often consider motion control along with sensors. Shan and Li et al. proposed an AV control strategy for path tracking, including curve fitting, curve discretization, and tracking controller design [9]. The Bessel curve fitting method was used to fit the generated results to create paths smoothly. Emmanuel proposed a fuzzy logic control strategy for AVs that combines GIS positioning, image processing, and image recognition [10]. Chen and He et al. proposed an innovative, hybrid path balancing model and established a virtual network to replace the original AV network. Each virtual link represents a group of paths connecting the entrance and the exit of the AV zone [11].

As can be seen, most of these studies did not focus on the requirements of the integrated traffic parameters. At present, the traffic index system of conventional road design is still adopted without considering the appearance of AVs or CVs. Taking the particularity of a vehicle network into account, this study intends to carry out a quantitative research on the integrated key traffic parameters, and put forward optimized running speed by balancing multiple objectives such as road efficiency, safety, and economy.

Moreover, scholars have made sufficient researches and discussions on the impact of AV on the traffic congestion, traffic safety, traffic flow theory, objective detection, and legal policies. Chen et al. explored the management of a fleet of shared autonomous electric vehicles (SAEVs) using a regional, discrete-time, and agent-based model [12]. The simulation examined the operation of SAEVs under various vehicle ranges and developed different scenarios in a city grid to analyze the impact of connected and autonomous vehicles (CAVs) on the infrastructure construction decisions. Fagnant et al. focused on the impact of CAVs on the traffic and environment. They established a city model that was divided into several grid regions, each of which were generated at a certain time interval. The variables such as the service area size, network congestion level, relocation strategy, and fleet size were analyzed successively. Thus, the impacts of CAVs on the environmental factors, such as urban travel efficiency, pollution emissions, and energy efficiency, were quantified [13]. The strategy designed in the study for exchanging CVs’ information is advisable: Vehicles share information, such as positions and speed, to adjust or reposition their own speed or acceleration according to the driving state of surrounding vehicles, so as to improve the traffic efficiency and optimize the overall running speed.

In regards to the traffic or road design parameters under the CAV environment, Chen et al. estimated the effects of autonomous trucks’ lateral distribution within the lane with respect to the rutting depth and fatigue damage by finite element analysis under certain environmental conditions [14]. Xie et al. combined VISSIM with the Car2X module (a built-in script in VISSIM that enables vehicles to exchange information) to accurately capture the acceleration, speed, and position of all vehicles in a certain area. The information was entered into an optimization model based on MATLAB, which then automatically decided the optimal control strategy for each vehicle and returned the simulation results [15]. In this paper, researches on running speed under the environment of instant traffic network were conducted, which requires a similar evaluation platform to compare and analyze the road traffic conditions at different running speeds. Based on the simulation results, the optimal traffic management strategy could be selected. Ntousakis et al. designed an analytical trajectory planning method to assist in the merging of vehicles on highways. It solved the problem of passengers’ discomfort by minimizing the acceleration [16]. In addition, an alternative solution to the time-varying linear quadratic regulator method was introduced, which applied the model predictive control (MPC) scheme to compensate for the possible interference in the trajectory of nearby vehicles. The results of the instant updated measurements were repeatedly applied to analyze the optimal solution until the merging process was finally completed [17]. This method was widely used in the area of the mixture of CVs and non-CVs. Ye and Yamamoto developed a heterogeneous traffic flow model to study the impact of connected vehicles on traffic flow [18]. Based on the recently proposed two-state safe-speed model, a two-lane cellular automaton model was developed, in which CAVs and conventional vehicles were incorporated into heterogeneous traffic flows [19]. To sum up, the relevant models of CAV have been explored, but there is little discussion on the intergraded traffic parameters.

Scholars have explored various micro-simulation models for AVs. However, it is difficult to verify the reliability of these models. In addition, the simulation methods for CVs still require further analyses and verifications. VISSIM is an effective platform for simulation and secondary development. It is mainly utilized for studying on-ramp path control, intersection signal optimization, and CAVs’ cruise control. Therefore, in this paper the traffic parameters in the context of CVs were defined and imported into VISSIM to conduct the simulation.

In summary, scholars have analyzed the impact of CAV driving technology in various fields on traffic flow. However, studies on the impact of CV technology on the basic and comprehensive traffic parameters are still rare. In an attempt to gain insight into the impact of vehicle networking on vehicles, this research simulated the environment of vehicle networking by applying the speed optimization model to the road model established in VISSIM. Besides, the influence of different CV rates on the results and the optimal CV marketing rate were also analyzed.

## 2. Model Development 

### 2.1. Road Model Based on VISSIM

Running speed refers to the safe speed that drivers can maintain under a fine road, traffic, and weather condition. A typical two-lane arterial road, of which the design speed is 50 km/h and the maximum speed is 70 km/h (nearly 20 m/s), was selected for VISSIM simulation. The parameter settings referred to the “Code for design of urban road engineering” of China [20] and the road situation (JiuZhu Road in Nanjing) was collected in-field for adjustment. The total length of the experimental road was 520 m and the lane width was 7.5 m without the median strip and shoulder. The road segments division is shown in Figure 1. The vehicle running speed, travel time, and delay time were obtained after the simulation.

Segment A, with a length of 200 m between points A and B, was used for data collection. Vehicles’ initial speed and positions were collected and entered into MATLAB in order to calculate the speed optimization model. Lane change will cause speed variations of vehicles (vehicles speed up obviously when changing lanes), which affects the accuracy of the initial speed. Therefore, it was assumed that vehicles cannot change lanes when passing through Segment A. Since Segment A was set for the initial data collection and not used for the traffic analysis, vehicles in segment A followed the VISSIM’s default Wiedemann 99 model.

Vehicles were allowed to change lanes in Segment B with a length of 200 m between points B and C, and the traffic flow eventually diverged into three directions at point C. In Segment B (and C), an average speed optimization model was adopted based on the available research and modified subject to the research objective to represent the vehicle’s connected state.

Segment C was the diversion zone of three directions, and the length of the main direction was 120 m. Göhring et al. proposed that the diversion ratio of traffic volume in the main direction and the other two diversion directions is 0.64:0.18:0.18 [21]. To verify the reliability of the ratio, a traffic survey on Caochang road in Nanjing was conducted. The actual diversion ratio was 6:2:2, which was consistent with the settings of Göhring et al [21].

Segment A was set for collecting the initial data, while segments B and C were designed for analyzing the connected state. Therefore, Segments B and C applied the average speed optimization model established in this paper, which will be discussed in Section 2.2. Running speeds were collected in these two segments. 

By default, a vehicle uses the entire width of the lane in VISSIM. The driving behavior and lateral behavior can be defined by the user. Otherwise the vehicles in a lane can drive on the left, on the right, or in the middle if a lateral orientation is not specified. If the overtaking vehicle is prevented from braking in time by the maximum deceleration, it should overtake another vehicle if possible. In this paper, CVs’ lane change rule (in Segments B and C) did not simply follow the default Widemann model. The parameters were modified through a separate experiment through VISSIM’s V2V sample. There is a V2V script in VISSIM, which is based on the real CV experiments. After applying this script to the road model, lane change parameters of higher credibility could be obtained. Thus, all driving behavior parameters for the lane change and lateral behavior could be modified in the Base Data menu in VISSIM when the optimization speed model was implemented afterwards. 

### 2.2. Optimization Speed Model Based on MATLAB

In order to simulate the connected state, an average speed optimization model was applied based on MATLAB, which was to obtain the maximum average value of running speed. This model was later combined with VISSIM to simulate the information exchanging process among the adjacent vehicles, which was regarded as the connected state. The VISSIM model without the application of the average speed optimization model was regarded as the normal state (vehicles do not exchange information).

#### 2.2.1. Optimal Speed Control Strategy

The findings of Xie et al. indicated that the optimal speed control strategy for CVs could be formulated as a nonlinear optimization [15]. The control strategy in their study was determined by a maximum total speed of all the vehicles passing through a certain area. However, the study was based on the assumption that the total number of vehicles was fixed, which is not sufficiently reasonable and practical. 

In this research an improved model was proposed, which took the maximum average speed as the objective. It is named as the average speed optimization model, as shown in Equations (1) to (6). The model adjusted Widemann 99 model by limiting CVs’ distances. Besides, the minimum total acceleration was added in this model as one of the decision factors because passengers do not want too much acceleration or deceleration in reality. It should be noted that if the maximum average speed and the minimum total acceleration could not be satisfied at the same time, the minimum total acceleration was considered to be smaller than the average values of all the situations that were calculated in advance.
(1)$$\left\{\begin{array}{l} \mathrm{Min}\frac{\left(-\sum_{i=1}^{n}\sum_{t=1}^{10}v_{i,t}\right)}{n}\\ \mathrm{Min}\sum_{i=1}^{n}\sum_{t=1}^{10}\left|a_{i,t}\right|\mathrm{or}\sum_{i=1}^{n}\sum_{t=1}^{10}\left|a_{i,t}\right|<C \end{array}\right.$$
(2)$$0<v_{i,t}<v_{\max},\forall i,t$$
(3)$$G_{\min}\leq \left|x_{i,t}-x_{i-1,t}\right|,\forall i,t$$
(4)$$a_{\min}\leq a_{i,t}\leq a_{\max},\forall i,t$$
(5)$$\frac{x_{i,t}-x_{i,t-1}}{t-(t-1)}=v_{i,t};\frac{v_{i,t}-v_{i,t-1}}{t-(t-1)}=a_{i,t}$$
*i* = vehicle number (*i* = 1, 2, 3…)*n* = total number of vehicles in Segment B*t* = the *t*th time step*a_i,t_* = acceleration of vehicle *i* at time step *t**v_i,t_* = velocity of vehicle *i* at time step *t**x_i,t_* = distance of vehicle *i* at time step *t* to the merging point*v_max_* = speed limit (here *v_max_* = 20 m/s; the design speed is 50 km/h)*G_min_* = minimum distance gap (here *G_min_* = 10 m, considering driving safety)*a_min_* = minimum acceleration rate (here *a_min_* = −5 m/s^2^, estimated based on the measured data)*a_max_* = maximum acceleration rate (here *a_max_* = −5 m/s^2^, estimated based on the measured data)*C* = the average total acceleration of all the situations, which is calculated in advance

This model took 10 s as a decision interval and was further divided into ten one-second decision steps. At the beginning of each one-second step, VISSIM passed the collected speed and position of each vehicle into MATLAB. Then the optimization model calculated the optimal control strategy and returned it to VISSIM for further control. The decision parameter was the acceleration (*a_i,t_*) of each vehicle. By optimizing these accelerations, the target of maximizing the average running speed was achieved.

#### 2.2.2. Using MATLAB to Implement Optimization Speed Model

In this paper, the Fmincon function in MATLAB (2015b, MathWorks Co. Ltd., Natick, MA, America) was utilized to conduct the iterative computation of the optimal average speed control strategy. Fmincon is a local optimization function that is designed to obtain the minimum value of the multivariate constrained nonlinear function.

The iterative method is shown in Figure 2.

Firstly, the main function was created. In this process, the initial values of speed, acceleration, and position for each time step were stored into a matrix and the limitations of speed and acceleration are set. Then the Fmincon optimization function was used for iteration so that the optimal solution was obtained.

Secondly, the nonlinear constraints were established. Two nonlinear constraints, Equations (2) and (3), were involved in this model, according to which the constraint function was constructed.

Next, the decision variable a (acceleration) was calculated by iterations. Functions were constructed to calculate the decision variable a, so that the average speed after 10 iterations was the largest, while meanwhile the total acceleration was reasonable.

Finally, the speed and distance were calculated. After the calculation of the acceleration, velocity and distance were calculated by the relationship of distance, velocity, and acceleration, and were finally stored in the matrix.

### 2.3. Simulation Framework

In order to establish the data connection between MATLAB and VISSIM, a C++ program was developed in this study to detect and output vehicle data in VISSIM, and Excel was adopted to transfer data between MATLAB and VISSIM.

The calling and control principle of the simulation model is shown in Figure 3. The simulation framework includes the following steps:1The road model and the traffic environment were established in VISSIM for micro simulation.2The experiment results were read in the VISSIM’s COM (Cluster Communication Port) interface. Specifically, the speed, acceleration, and position of the vehicles in Segment A were read per second. Such an advanced simulation was handled through the VISSIM–COM interface and implemented in MATLAB.3Excel was adopted as the intermediate program to realize data transmission of VISSIM and MATLAB. MATLAB was called through the Excel link interface, and the simulation data was transferred between VISSIM and MATLAB.4The MATLAB optimization speed model was adopted with the Fmincon algorithm to optimize the average running speed. Then MATLAB returned the output value per second (i.e., the acceleration), and stored it in Excel via Excel link [22].5The optimization decision was read in Excel through the COM interface, and the COM interface was connected with Excel. A C++ program was written to control the vehicles as the kernel of the model [23].6The instant simulation of VISSIM was based on the optimized decision, by which the vehicle exchanged information instantly under the connected state and determines its own speed and acceleration. Then VISSIM output the simulation evaluation data.

## 3. Results

### 3.1. Model Verification

To demonstrate how the optimization-based control algorithm works, the initial values (as shown in Table 1) were manually entered to verify the validity of the proposed model. Since vehicles were not allowed to change lanes in Segment A, the following two cases only considered one lane respectively.

#### 3.1.1. Case 1

In this case, the position and entry speed of each vehicle were intercepted and imported corresponding to the vehicle number. It was assumed that the initial acceleration was zero. As shown in Table 1, data were collected at one time when VISSIM had run for a sufficient time to ensure the randomness (without applying the average speed optimization model). 

Speed after optimization is shown in Figure 4. It can be found that the speed of each vehicle was close to the maximum value after several iterations, and all vehicles traveled at the same speed in the last 2 s. Except for vehicle 272, all vehicles basically reached the maximum value in the third iteration. However, due to the minimum distance limitation, vehicle 272 slowed its acceleration, and reached the maximum speed actually in the 8th iteration, which proved the minimum distance limitation worked. 

Distance of vehicles from point C is shown in Figure 5. It can be found that the six vehicles maintained a safe distance of more than 10 m during 10 iterations. It can be concluded that the average speed optimization model was verified under this condition. In addition, vehicles could move at a speed close to the limited maximum speed while maintaining a safe distance, and achieved the goal of improving road capacity.

#### 3.1.2. Case 2

The second case was carried out to prove that both lanes in Segment A were applicable to the average speed optimization model, and also to verify the average speed optimization model. The initial information is shown in Table 2.

Speed after optimization on lane 2 is shown in Figure 6. The speeds of nine vehicles were close to the maximum value at the fourth iteration, and all vehicles were running at a constant speed after the fifth iteration. From Figure 7, it can be seen that there were small changes in the distances between vehicles in the first 5 s. In the last 5 s, the curves were basically nine parallel straight lines. The distances between adjacent vehicles were greater than the safe distance of 10 m, which also verified the optimization speed model.

### 3.2. Speed Optimization Results

#### 3.2.1. Analysis on Data Collection Points

Data collection points were set on the two lanes at points B and C in Figure 1, respectively. The collection points at point B on lanes 1 and 2 were named as collection points 1 and 2, and the collection points at point C on lanes 1 and 2 were named as collection points 3 and 4. The average speed and acceleration of the vehicles at each collection point are shown in Table 3. The connected state was the situation when VISSIM was combined with the average speed optimization model and the normal state was the situation when VISSIM directly ran without using the model.

In Table 3, the average speed in the connected state was larger than that in the normal state at every point, and the acceleration in the connected state was smaller. Moreover, with the objective to optimize average speed, vehicles approached a constant speed when their distances reached the minimum value. In other words, the acceleration of CVs was close to zero. Similarly, the acceleration at point C was higher than that at point B and most vehicles were accelerating.

In Table 4, the speed deviation of collection point 4 was larger than that of collection point 2, and the maximum acceleration of collection point 2 was 2.00 m/s^2^, which did not exceed the acceleration limitation of *a_max_* = 5m/s^2^. Moreover, the minimum acceleration and the maximum speed did not exceed the limitation. This shows that under the average speed optimization model, the limits on the speed, acceleration and minimum distance could be satisfied. The data at collection points 1 and 3 had the similar tendency as that at collection points 2 and 4, so it is not shown in this paper.

It can be seen that the average vehicle speed in the connected state was about 4 km/h higher than that in the unconnected state.

In addition, in Table 3, the acceleration at collection point 1 did not change much in both cases, and the acceleration at collection point 2 increased at a large extent, from −0.4 m/s^2^ to nearly zero. The reason is that the average speed optimization model sought the optimal value in the process of speed optimization. The vehicles ran at a constant speed, when their distances reached the minimum and the total speed reached the maximum after multiple iterations.

#### 3.2.2. Analysis on Travel Time and Delay Time

Figure 8 and Figure 9 express the average travel time and delay time of vehicles in the two states, respectively. It can be seen that the average travel time and delay time of CVs in every period were shorter than those in the normal state.

According to Figure 8 and Figure 9, in the connected state, the average travel time and delay time of each time period in segments A and B were reduced. The total travel time and delay time decreased by 5.34% and 16.76%, respectively. The main reason is that the overall speed of the vehicles was accelerated by the average speed optimization model, and thus the traffic flow became smoother.

#### 3.2.3. Analysis of the Mixed Traffic Flow in the Normal and the Connected States

In order to analyze the influence of CV market penetration rates on the results, the two states should be considered together. 

To realize the mixed traffic flow, two kinds of vehicles were set up in VISSIM. The rate of different kinds of vehicles could be directly set by entering different vehicle volumes in VISSIM. Keeping the total traffic volume constant, a group of comparative experiments were conducted, with CV rates ranging from 0 to 1 (0, 0.2, 0.4, 0.6, 0.8, 1). Rates 0 and 1 represent the conditions in Section 3.2.1 and Section 3.2.2.

Two classes of vehicles were defined in MATLAB, and the optimization speed model was only implemented for one class (CV). Then the new code was run, and the results of the mixed traffic flow were obtained.

Figure 10, Figure 11 and Figure 12 reflect the average running speed, travel time, and delay time of the mixed traffic flow, respectively. The data was collected from 500 to 600 s in Segment 2, when the simulation had run for a sufficient duration.

From Figure 10, Figure 11 and Figure 12, it can be seen that there was an obvious descending or ascending trend on the average running speed, travel time, and delay time when the CV rate increased, which indicates the CV rate would remarkably influence the results. The trend of the average travel time was very steady and close to a straight line, while there were leaps in the trend of the other two figures at the rate of 0.6. It can be concluded that the running speed and delay time were significantly improved when the CV rate is 0.6, which was supposed to be the optimal CV market penetrating rate.

## 4. Discussion

In this paper VISSIM was utilized to build the road model for CVs simulation. The results of this study were based on simulation, but they provide a new orientation for studies of CVs: Statistical analysis before implementing CV experiments in real roads, which is more economical and can provide estimations for road design and traffic analysis, etc.

Results quantitatively showed how vehicle networking affects CVs’ running speed, travel time, and delay time. Although the results of this study depended on different kinds of software, they provide a reference for the study of safety and efficiency of CVs. When the model was verified, it was found that the running speed tends to be stable after several iterations. This is consistent with Xie’s findings [15].

Although Fmincon solves nonlinear constraints, the objective function should be continuous. The function decided step size according to the given initial value, and could only obtain the local optimal solution. As was mentioned by Voelz, if the initial value given was an infeasible solution, it was difficult to converge to the optimal solution, and the final result might exceed the upper and lower limits in order to reach the target function value [24]. It must be pointed out that the average speed optimization model in this research was not suitable for the crowded traffic flow. The traffic flow of the simulation model designed in this paper was basically unobstructed, with only little queuing of vehicles. Some concerns need to be clarified:1.Most vehicles’ speed in the crowded traffic flow tended to be zero, which brought negative effects on the accuracy of speed analysis.2.The vehicle distance was limited to 10 m in this model, but it was not true when the traffic was blocked. As a result, it is not conductive to analyzing queue length and vehicles’ passing time.

Under these concerns, the crowded traffic situation was not considered in this study. It should be noted that, till now, CAVs are mostly experimented in highways or areas with a good traffic condition, where the results of this study are applicable. As for the traffic flow in the crowded state, other speed optimization models need to be considered, such as ant colony optimization [25], genetic algorithm [26], and so on.

Based on the comparative analysis of running speed in the normal and the connected states, it can be concluded that the average running speed of the vehicle in the connoted state could increase by about 4 km/h compared to that in the normal state. Considering the influence of various other factors on the real road and the hardware and software limitations in the simulation process, running speed in the connected state is proposed: In the case of fully-connected driving conditions, the running speed of urban arterial roads with bidirectional diversions with a maximum speed limit of 70 km/h can increase by 4 km/h. However, other factors still need to be considered to analyze the running speed under actual vehicle networking conditions, such as the safety when the vehicle turns at a corner. A comprehensive study of the minimum radius of the road is required. Recently, there are many kinds of automatic control algorithms for CAVs, and errors in simulations are inevitable. Therefore, these issues need to be considered for the precise adjustment of running speed comprehensively. There was only one road model in this study, but this kind of road is very common in China. It is hoped that more samples can be supplemented in future studies. The road model was conducted according to Chinese roads, but the internal design parameters in VISSIM, such as the lane change and lateral behavior, were modified through a separate experiment through the VISSIM’s V2V (vehicle to vehicle) sample to ensure the credibility of the results. The driver responses of CVs were ignored, as it is assumed that the vehicle was under a totally connected state without human control. Above all, the method mentioned in this paper provides a simulation-based adjustment scheme for running speed, which can be used for preliminary estimation and reference before AV experiments. 

## 5. Conclusions

In order to study the impact of connected driving state on the vehicle running speed, an optimization speed control strategy was developed in MATLAB and combined with VISSIM simulation. Based on various constraints on the maximum average running speed, the acceleration, distance, and running speed in the simulation were optimized. The optimization of running speed was realized by the optimization toolbox in MATLAB, so as to simulate the traffic flow in both the normal and the connected states. After that, the impacts of the connected state on the running speed, travel time, and delay time were captured. Taking the CV penetration rate into consideration, this study compared the effects of different CV rates on the mentioned traffic characteristics. This study further established a multi-index traffic evaluation method for running speed and analyzed the impacts of the mixed CV flow scenario. The conclusions of this paper are as follows:(1)The running speed in the connected state is 4 km/h larger than that in the normal state. It proves that the fully-connected environment can improve running speed significantly.(2)The average total travel time and delay time of changing lanes or diverging decrease by 5.34% and 16.76% in the connected state, respectively. It shows that the fully-connected environment can make good use of the existing road infrastructure and improve traffic efficiency.(3)The optimal CV market penetrating rate is 0.6. At such a rate, the running speed, travel time, and delay time is 56.21 km/h, 14.77 s, and 0.76 s, respectively.

## Figures and Tables

**Figure 1 ijerph-16-04373-f001:**
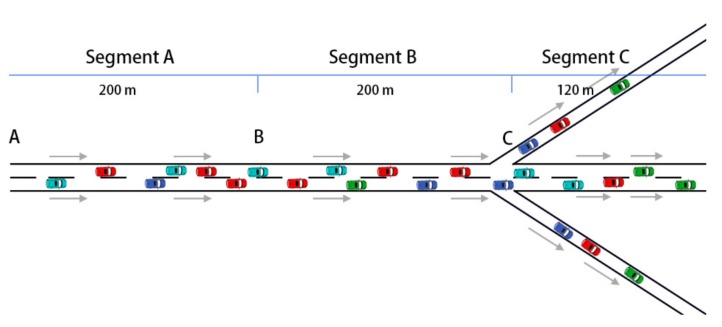
Road segments division.

**Figure 2 ijerph-16-04373-f002:**
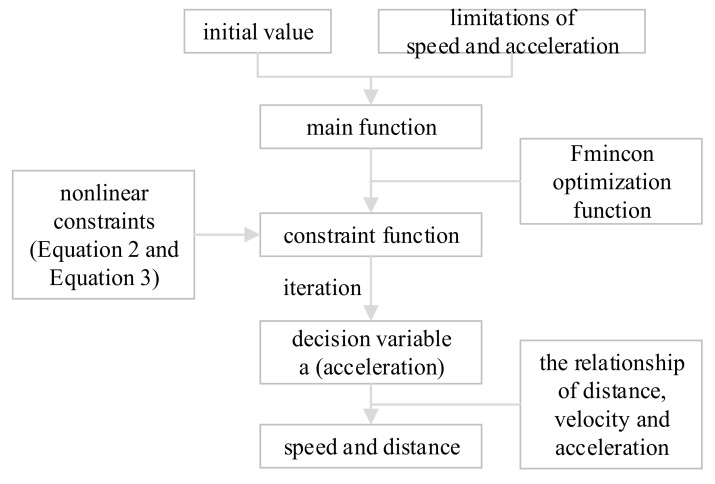
Iterative method.

**Figure 3 ijerph-16-04373-f003:**
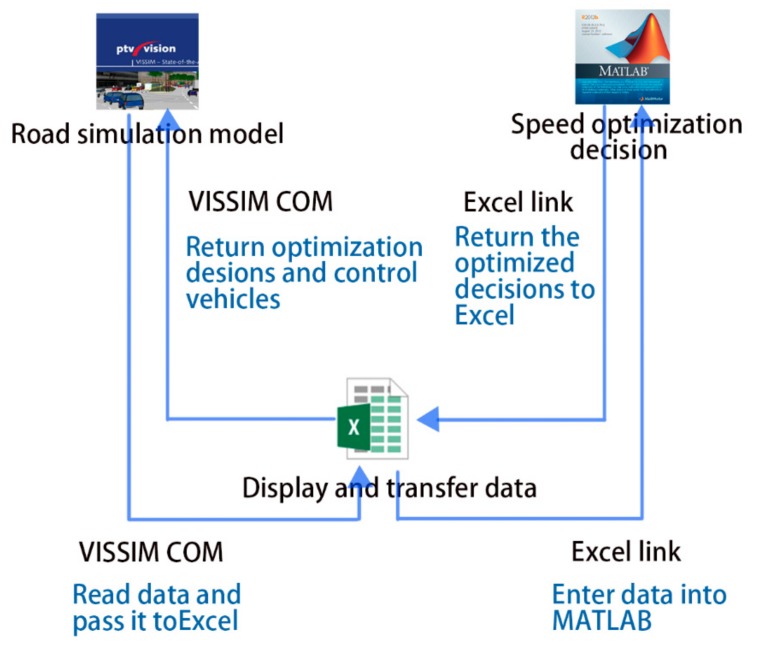
Principle of calling and controlling of the simulation model.

**Figure 4 ijerph-16-04373-f004:**
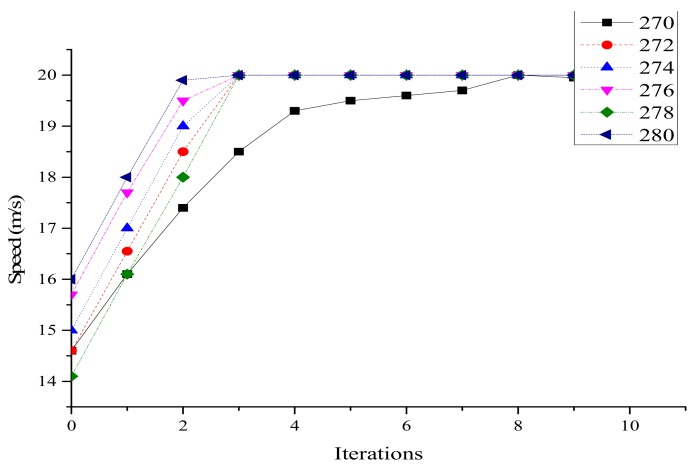
Optimized vehicle speed on lane 1.

**Figure 5 ijerph-16-04373-f005:**
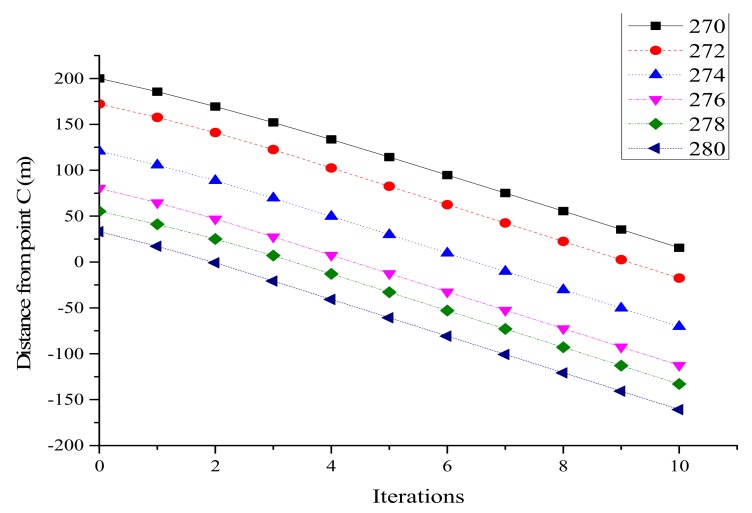
Distance from point C on lane 1.

**Figure 6 ijerph-16-04373-f006:**
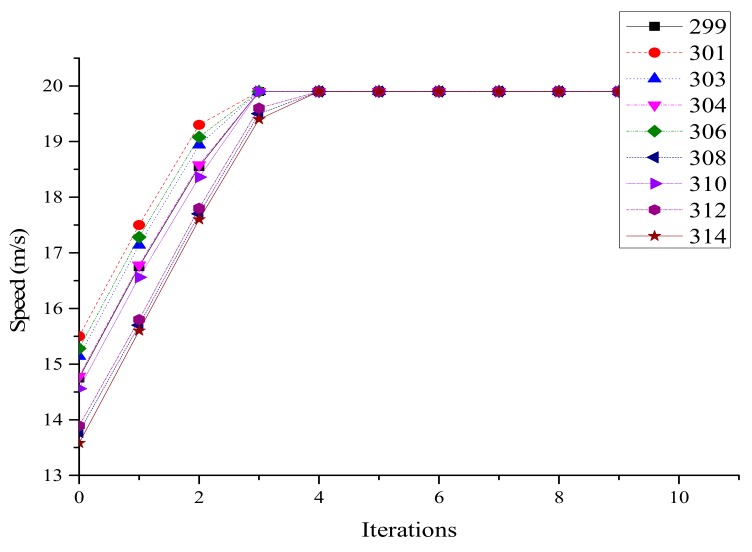
Optimized vehicle speed on lane 2.

**Figure 7 ijerph-16-04373-f007:**
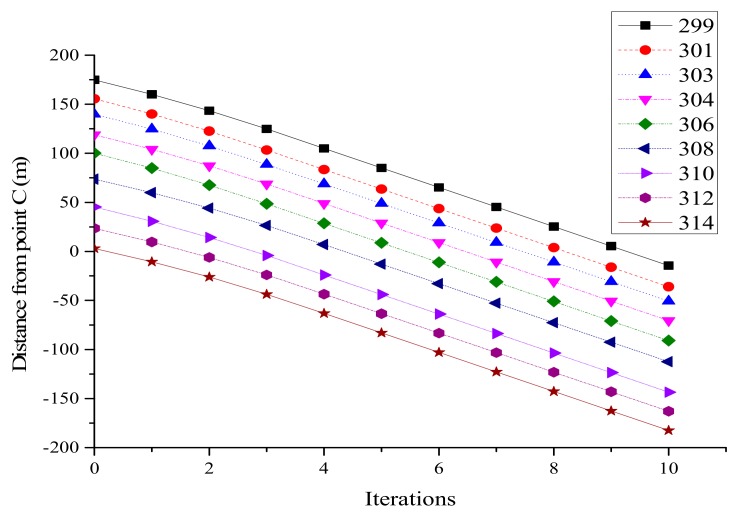
Distance from point C on lane 2.

**Figure 8 ijerph-16-04373-f008:**
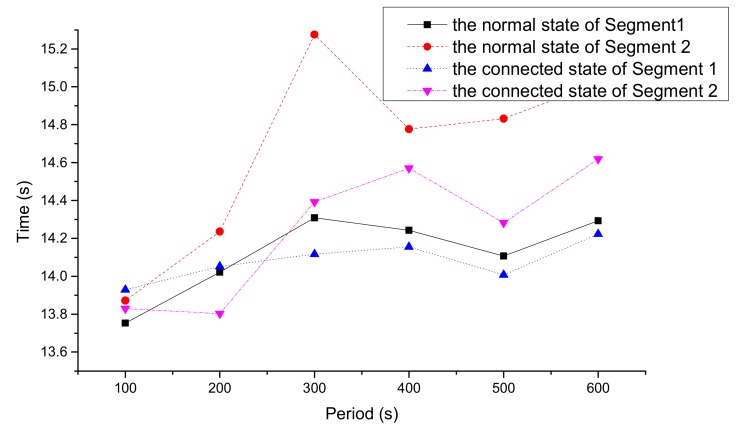
Average travel time in two states.

**Figure 9 ijerph-16-04373-f009:**
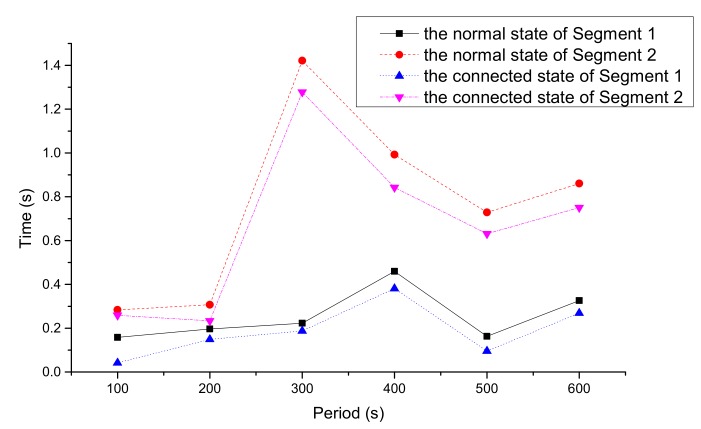
Average delay time in two states.

**Figure 10 ijerph-16-04373-f010:**
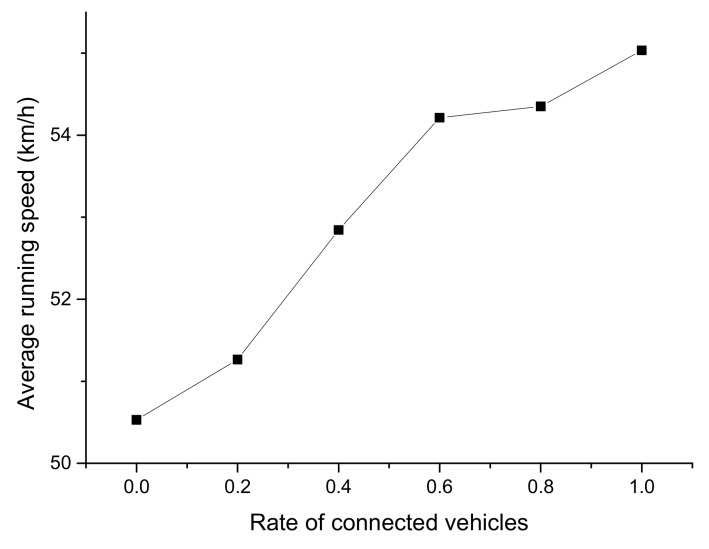
Average running speed from 500 to 600 s with different CV rates.

**Figure 11 ijerph-16-04373-f011:**
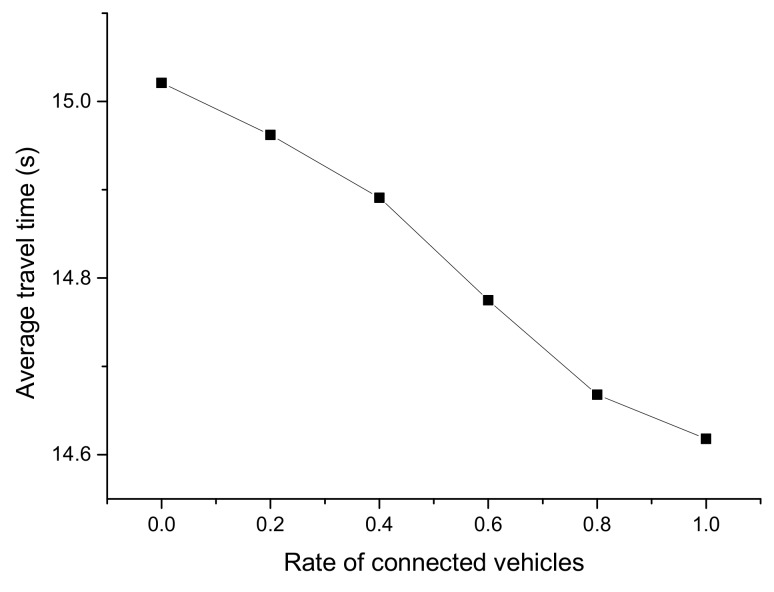
Average travel time from 500 to 600 s with different CV rates.

**Figure 12 ijerph-16-04373-f012:**
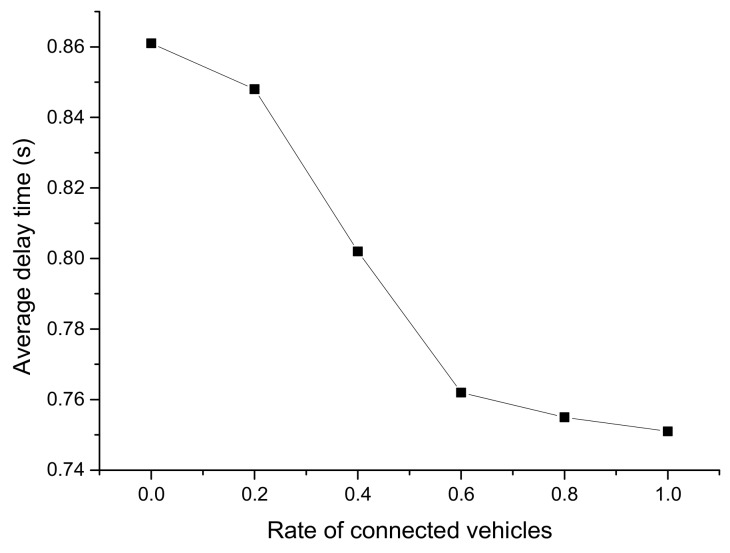
Average delay time from 500 to 600 s with different CV rates.

**Table 1 ijerph-16-04373-t001:** Initial speed information on lane 1.

Time(s)	Number	Position	Speed (km/h)	Speed (m/s)
500	321	200.13	52.56	14.6
500	322	172.16	52.56	14.6
500	323	120.68	54	15
500	324	80.36	56.52	15.7
500	325	55.23	50.76	14.1
500	326	33.12	57.6	16

Number: The code of one vehicle. Position: The vehicle’s distance from point C.

**Table 2 ijerph-16-04373-t002:** Initial speed information on lane 2.

Time(s)	Number	Position (m)	Speed (km/h)	Speed (m/s)
568	299	174.89	53.1	14.75
568	301	155.63	55.8	15.50
568	303	139.87	54.5	15.14
568	304	118.87	53.2	14.78
568	306	100.13	55.0	15.28
568	308	73.65	49.6	13.78
568	310	45.27	52.4	14.56
568	312	23.52	50.0	13.89
568	314	3.00	48.9	13.58

**Table 3 ijerph-16-04373-t003:** Average speed and acceleration in two cases.

Collection Point	The Connected State		The Normal State	
	Speed (km/h)	Acceleration (m/s^2^)	Speed (km/h)	Acceleration (m/s^2^)
1	55.412	−0.078	51.298	−0.102
2	55.035	−0.025	50.529	−0.029
3	55.4	0.255	51.387	0.26
4	55.659	0.26	51.608	0.246

**Table 4 ijerph-16-04373-t004:** Speed and acceleration of vehicles at collection points 2 and 4.

	Collection Points		Average	Maximum	Minimum	Standard Deviation
The normal state	2	Speed (km/h)	50.529	60.074	43.573	4.093
		Acceleration (m/s^2^)	−0.029	2.016	−6.933	0.689
	4	Speed (km/h)	51.608	59.619	28.177	4.927
		Acceleration (m/s^2^)	0.246	2.398	−0.322	0.559
The connected state	2	Speed (km/h)	55.035	62.426	47.188	3.908
		Acceleration (m/s^2^)	−0.025	2.009	−1.921	0.874
	4	Speed (km/h)	55.659	62.744	33.155	4.227
		Acceleration (m/s^2^)	0.26	2.408	−0.314	0.564

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
