# Peer review of "Traffic Simulation Analysis on Running Speed in a Connected Vehicles Environment"

_ijerph, 2019, doi:10.3390/ijerph16224373_

Round 1

Reviewer 1 Report

Thanks for incorporating the suggestions.

Author Response

Thank you for the reviews. Please don’t hesitate to contact me if there are any other questions.

Reviewer 2 Report

Changes have been noted.  However, the paper title still seems problematic.  "Traffic Characteristic Analysis" doesn't seem to be an appropriate word to describe the research content.  The authors should consider the use of terms such as traffic simulation, ...etc, in the title.  

The authors should give a proper definition of "running speed" in terms of connected vehicles in the abstract, and in the paper, to help readers better understand the content.

The amendment in line 111 doesn't state clearly the simulated road segment is based on a physical road segment,i.e., "previous road situations" doesn't convey this meaning.  State the actual road name, ...etc, better.

Some typo- and grammatical errors still exist, e.g., line 14, "In two-lane arterial road" should be "In a two-lane arterial road".  The authors should really check and correct all similar errors thoroughly again.  

The use of references is also questionable.  For example, in one of the key references, 19, Gohring et al., not being the authors, was quoted as proposing the split ratio.  If this is the case, why not directly quote the publication by Gohring et al.?  I think the use of the particular ratio in the paper has not been well explained.  In terms of traffic analysis, the directional split of the three directions should depend upon a number of factors, and there is no golden rule of how the traffic should split, and if it's purely an assumption, then the authors should better explain clearly why and how to come up with these values.  Why aren't they varied in the experiment?  

Furthermore, the authors have used a number of assumptions, e.g., lane-changing policy (lines 116 to 119), which may not be a valid assumption in micro-simulation experiments, not even in a connected vehicles' environment.  There are many possible variation of factors and environmental setting that can affect the experiments.  The authors by placing rigid assumptions on their experiments fail to establish the creditability of their model and results.  What's mentioned in lines 332 to 338 are not sufficient to explain the limitation of this study.  The authors should go one step further to overcome this limitation to really prove and build credibility of their results.

Author Response

Response to Reviewer 2 Comments

Thank you very much for the reviews. Below is the explanation for the question respectively. Don’t hesitate to contact me if there are any other concerns.

Point 1: "Traffic Characteristic Analysis" doesn't seem to be an appropriate word to describe the research content.  The authors should consider the use of terms such as traffic simulation, ... etc, in the title.  

Response 1: We have changed the title to “Traffic Simulation Analysis in a Connected Vehicles Environment on Running Speed” to better describe the content.

Point 2: The authors should give a proper definition of "running speed" in terms of connected vehicles in the abstract, and in the paper, to help readers better understand the content.

Response 2: We have added the explanation of running speed in Section 2.1.

Pont 3: The amendment in line 111 doesn't state clearly the simulated road segment is based on a physical road segment,i.e., "previous road situations" doesn't convey this meaning.  State the actual road name, ...etc, better.

Response 3: We have modified the statement in Section 2.1 to “The parameter settings refer to "Design Standard of Urban Road Planning " of China and take actual road situation (JiuZhu Road in Nanjing) for adjustments.”

Pont 4: Some typo- and grammatical errors still exist, e.g., line 14, "In two-lane arterial road" should be "In a two-lane arterial road".  The authors should really check and correct all similar errors thoroughly again.

Response 4: We have checked the mistakes and corrected grammatic errors through the paper.

Pont 5: The use of references is also questionable.  For example, in one of the key references, 19, Gohring et al., not being the authors, was quoted as proposing the split ratio.  If this is the case, why not directly quote the publication by Gohring et al.?

Response 5: Reference [19] is changed to the publication by Gohring et al.

Pont 6: I think the use of the particular ratio in the paper has not been well explained. In terms of traffic analysis, the directional split of the three directions should depend upon a number of factors, and there is no golden rule of how the traffic should split, and if it's purely an assumption, then the authors should better explain clearly why and how to come up with these values.

Response 6: To verify the reliability of the ratio, a traffic survey on Caochang road in Nanjing has been conducted. The actual diversion ratio is calculated to be about 6:2:2, which is basically consistent with the study.

Pont 7: Furthermore, the authors have used a number of assumptions, e.g., lane-changing policy (lines 116 to 119), which may not be a valid assumption in micro-simulation experiments, not even in a connected vehicles' environment. 

Response 7: Segment A is just set for collecting initial data in order to calculate the speed optimization model (this explanation has been added into section 2.1 to make it clearer that Segment A is not for analysing vehicle connected state), the actual segments for analysing connected state are only Segment B and C. Lane change will cause number and speed variations of vehicles (vehicles speed up obviously when changing lanes), which affects the accuracy of initial speed. Therefore, it is assumed that vehicles can’t change their lanes when passing through Segment A.

Pont 8: There are many possible variations of factors and environmental setting that can affect the experiments. The authors by placing rigid assumptions on their experiments fail to establish the creditability of their model and results. What's mentioned in lines 332 to 338 are not sufficient to explain the limitation of this study. The authors should go one step further to overcome this limitation to really prove and build credibility of their results.

Response 8: There is only one road model in this study, but this kind of road is very common in China. The following improvements have been made to prove credibility of the results:

We have added explanations for the user-defined lane change and lateral behaviour rules for the model, and the credibility of these parameters in Section 2.1. We have added supplementary experiments for the influence of different CV rate on the results in Section 3.2.3, which is more universal for actual conditions.

Reviewer 3 Report

I'm in concept what you are trying to show, I don't understand how it would be implemented in practice.  What information is exchanged between vehicles (I'm assuming BSM data)?  How does the desired speed get communicated the the vehicles in practice?  How do the results change as more traffic volumes is introduced to the system?  What are the effects of CV market penetration rates of resutls?

Author Response

Response to Reviewer 3 Comments

Thank you very much for the reviews. Below is the explanation for the questions respectively. Don’t hesitate to contact me if there are any other concerns.

Point 1: I don't understand how it would be implemented in practice. 

Response 1: There are several scenarios where it would be implemented in practice:

The recommended speed limit for urban road when vehicles are connected. Delay prediction under vehicle connected state, according to which transportation facilities (e.g. traffic lights) and management strategy could be adjusted. Road design index analysis, like design speed and any other index related with running speed.

Point 2: What information is exchanged between vehicles (I'm assuming BSM data)?

Response 2: Each vehicle’s position, speed and acceleration are exchanged.

Pont 3: How does the desired speed get communicated the vehicles in practice?

Response 3: There are kinds of ways for vehicles to communicate and share their travel information in real condition: VAD (e.g. radio), ASD (It can not only send current vehicle safety message to remote vehicles, but also receive basic safety message from distant vehicles.), or RSD. In the simulation, the desired speed is calculated in MATLAB by vehicles’ real-time position and speed, and is put into VISSIM every 1 second for further control.

Pont 4: How do the results change as more traffic volumes is introduced to the system?

Response 4: As discussed in the 3rd paragraph in Discussion section, the optimization average speed model in this research is not suitable for the crowded traffic flow. This is based on the below concerns:

Most vehicles’ speed in crowed traffic flow tends to be 0, which will badly affect the accuracy of speed analysis. The vehicle distance is limited in 10m in this model, but it is not true when traffic is blocked up. As a result, it is not conductive to analyze queue length and vehicle passing time.

Under these concerns, the crowded traffic situation is not considered. It should be noted that till now connected and autonomous vehicles are mostly experimented in highway or areas with good traffic condition, where the results of this study are applicable.

Pont 5: What are the effects of CV market penetration rates of results?

Response 5: We have added supplementary experiments for the influence of different CV rates on the results in section 3.2.3, and added the relevant conclusion.

Reviewer 4 Report

Abstract should be improved and clarified so that the salient features of this research are better brought up to the readers.

Author Response

Dear Reviewer:

Thank you very much for the review. According to the suggestion, we adjust the abstract by adding details into it:

Brief description for the speed optimization model. Supplementary descriptions for the content of the result. Practical significance of the study.

Don’t hesitate to contact me if there are any other concerns.

Best Regards

Round 2

Reviewer 2 Report

The English writing still has rooms to improve.  For example: line 16 - "Conclusions : ...", a complete sentence is expected in the abstract.  Lines145 to 149, "If the maximum deceleration prevents the overtaking vehicle from braking in time, it overtakes ...the driving behavior parameters Consider next...", which is one of the examples in the paper of a poorly written sentence with hard to follow logic.  The English must really be improved before this paper can be published.  

Author Response

Response to Reviewer 2 Comments

Point 1: The English writing still has rooms to improve.

Response 1: Thank you for your comments and suggestions very much. The two points you mentioned have been modified as below. Besides, we have checked the errors throughout the paper and marked yellow.

Point 2: line 16 - "Conclusions : ...", a complete sentence is expected in the abstract.

Response 2: We have polished this sentence in the abstract.

Point 3: Lines145 to 149, "If the maximum deceleration prevents the overtaking vehicle from braking in time, it overtakes ...the driving behavior parameters Consider next...".

Response 3: We have modified this sentence to “If the overtaking vehicle is prevented from braking in time by the maximum deceleration, it should overtake another vehicle if possible.”